# Tear Film Amphiphilic and Anti-Inflammatory Lipids in Bovine Pink Eye

**DOI:** 10.3390/metabo8040081

**Published:** 2018-11-21

**Authors:** Paul L. Wood, Michelle N. Donohue, John E. Cebak, Taylor G. Beckmann, MacKenzie Treece, Jason W. Johnson, Lynda M. J. Miller

**Affiliations:** 1Metabolomics Unit, College of Veterinary Medicine, Lincoln Memorial University, 6965 Cumberland Gap Pkwy, Harrogate TN 37752, UK; michelle.donohue@lmunet.edu (M.N.D.); john.cebak@lmunet.edu (J.E.C.); mackenzie.treece@lmunet.edu (M.T.); 2Department of Medicine, DeBusk College of Osteopathic Medicine, Lincoln Memorial University, 6965 Cumberland Gap Pkwy, Harrogate TN 37752, UK; taylor.beckmann@lmunet.edu; 3College of Veterinary Medicine, Lincoln Memorial University, 6965 Cumberland Gap Pkwy, Harrogate TN 37752, UK; Jason.johnson@lmunet.edu (J.W.J.); lynda.miller@lmunet.edu (L.M.J.M.)

**Keywords:** bovine pink eye, infectious bovine keratoconjunctivitis (IBK), resolvin E2, cyclic phosphatidic acid, (O-acyl)-ω-hydroxy-fatty acids, plasmalogens, sphingomyelins, ocular inflammation, tear film fluid

## Abstract

*Background*: Tear film fluid serves as a dynamic barrier that both lubricates the eye and protects against allergens and infectious agents. However, a detailed analysis of a bacteria-induced immune response on the tear film lipidome has not been undertaken. *Methods*: We undertook a high-resolution mass spectrometry lipidomics analysis of endogenous anti-inflammatory and structural tear film lipids in bovine pink eye. *Results*: Bovine pink eye resulted in dramatic elevations in tear fluid levels of the anti-inflammatory lipids resolvin E2, cyclic phosphatidic acid 16:0, and cyclic phosphatidic acid 18:0. In addition, there were elevated levels of the structural lipids (O-acyl)-ω-hydroxy-fatty acids, cholesterol sulfate, ethanolamine plasmalogens, and sphingomyelins. Lipid peroxidation also was augmented in pink eye as evidenced by the hydroperoxy derivatives of ethanolamine plasmalogens. *Conclusions*: Ocular infections with *Moraxella bovis* result in the induction of a number of endogenous anti-inflammatory lipids and augmentation of the levels of structural glycerophospholipids and sphingolipids. Increased levels of hydroperoxy glycerophospholipids also indicate that this bacterial infection results in lipid peroxidation.

## 1. Introduction

Immune responses drive the innate inflammatory response that acts to limit harmful pathogens and infections. However, a sustained immune response can ultimately lead to cellular dysfunction and cell death [1,2,3]. Resolution of ongoing inflammatory responses is complex and has been shown to involve a number endogenous pro-resolving lipids which include the lipoxins, resolvins, protectins, and maresins [4,5]. In addition cyclic phosphatidic acids [6,7] and very-long-chain dicarboxylic acids [8] are endogenous anti-inflammatory lipids. A detailed study of these lipids has not been undertaken in ocular inflammation. In this regard, bovine infectious keratoconjunctivitis is an excellent model to evaluate endogenous anti-inflammatory responses that are initiated to resolve the initial immune response. *Moraxella bovis* is a gram-negative coccobacillus infection that results in lacrimation, photophobia, and blepharospasm. Corneal edema and ulcers develop with re-epithelialization of the corneal ulcers occurring in 2 to 3 weeks. In this regard, resolvin E1 has been demonstrated to reduce inflammation in a murine ocular model of bacterial keratitis [9]. Both resolvin E1 and E2 are metabolites of eicosapentaenoic acid, with resolvin E2 also demonstrating potent anti-inflammatory activity [4,10]. These anti-inflammatory actions include suppression of neutrophil infiltration, suppression of cytokine production, and enhanced macrophage phagocytosis of cellular debris and microbial clearance. Similarly, very-long-chain dicarboxylic acids are endogenous anti-inflammatory lipids that suppress cytokine and nitric oxide production by monocytes [8,11]. Structural tear film lipids are also essential in maintaining the tear film structure to protect the eye against dehydration, allergens, and infection. The impact of bacterial infection on critical tear film structural lipids remains to be investigated. In this regard, (O-acyl)-ω-hydroxy-fatty acids [12], glycerophospholipids and sphingomyelins [13] are essential to the ultrastructure of the tear film. In summary, a detailed analysis of bacterial infection on the tear fluid lipidome has not been conducted to-date. Our study provides the first important step to these analyses.

## 2. Results

### 2.1. Anti-Inflammatory Lipids

Since many of the anti-inflammatory lipids have a number of isobars, we utilized MS^2^ to measure the individual mediators. The parent ion was isolated with a window of 1.0 amu while the product ion was acquired with high resolution (<2 ppm mass error). Utilizing this approach, the only anti-inflammatory lipids that were reliably monitored in both control and infected cattle were resolvin E2, cyclic phosphatidic acids, and very-long-chain dicarboxylic acids.

Pinkeye elicited a rapid induction of several of these key anti-inflammatory lipids. This included cyclic phosphatidic acids and resolvin E2, but not very-long-chain dicarboxylic acids (Figure 1). While cyclic phosphatidic acids were augmented, lysophosphatidic acid a lipid mediator with an opposing pharmacodynamic profile to that of cyclic phosphatidic acids [14], was unaltered (Figure 1). There were no differences in these lipid metabolites based on the stage of bacterial infection, suggesting that the host immune response occurs early after infection and is sustained.

### 2.2. Amphiphilic Lipids.

Both (O-acyl)-ω-hydroxy-fatty acids (OAHFA) and cholesterol sulfate were significantly increased in the tear film of infected cattle (Figure 2). The predominant OAHFAs were OAHFA 48:2 and OHFA 50:2 The fatty acid constituents (fatty acid/ ω-hydroxy-fatty acid) of these lipids were determined by MS^2^ (<2 ppm mass error). Each of the OAHFAs had several isobaric forms with OHFA 48:2 mass including OAHFA 16:1/32:1(OH) and OAHFA 18:1/30:1(OH) while the mass for OHFA 50:2 included OAHFA 18:1/32:1(OH) and OAHFA 16:1/34:1(OH), similar to our previous observations with equine sperm [15] and amniotic fluid [16]. Cholesterol sulfate was also validated by MS^2^ with the product anion of HSO_4_ (96.9595; 0.82 ppm).

### 2.3. Structural Glycerophospholipids

A number of ethanolamine plasmalogens were increased in the tear film of infected cows (Figure 3). In addition, hydroperoxy derivatives of ethanolamine plasmalogens were detected (Figure 3), supporting ongoing lipid peroxidation in this ocular infection. The hydroperoxy lipids were characterized by the loss of H_2_O and H_2_O_2_ as previously reported for lipid peroxidation products in the serum of equine leptospirosis and leptospira-vaccinated horses [17].

### 2.4. Structural Sphingolipids

Several key sphingomyelins also were increased in the tear film of infected cows (Figure 4), similar to increases in serum sphingomyelins we have reported for the immune response to leptospira vaccination in horses [17].

## 3. Discussion

The ocular surface represents a large surface area that is exposed to the environment requiring hydration and lubrication as well as protection against potential insults from allergens and bacteria. The lipid layer is the outer layer of the tear film and is directly exposed to the external environment. This layer is composed of non-polar lipids (cholesterol esters, wax esters, triacylglycerols and diacylglycerols) with an inner amphiphilic layer of glycerophospholipids, (O-acyl)-ω-hydroxy-fatty acids, and cholesterol sulfate [12,13]. This amphiphilic layer provides the physical interaction of the outer non-polar lipid layer and the aqueous sublayer [12]. As amphiphiles, (O-acyl)-ω-hydroxy-fatty acids, sphingomyelins, glycerophospholipids [12,14,15] as well as cholesterol sulfate [15] may function as ocular surfactants which allow the interaction of the lipid and aqueous layers of the tear film and also may be bacteriostatic.

The lipids of the tear film come from the Meibomian glands [18,19,20,21]. The biosynthetic pathway for (O-acyl)-ω-hydroxy-fatty acids first involves the generation of very-long-chain fatty acids via ELOVL4 (elongation of very long chain fatty acids-4), an enzyme present in Meibomian glands [22]. The generation of ω-hydroxy-fatty acids by (CYP4A/F subfamilies) to generate the ω-hydroxy function and the acyltransferase(s) involved in the ultimate formation of the OAHFA products remain to be characterized in Meibomian glands. (O-acyl)-ω-Hydroxy-fatty acids have also been shown to serve complex lipid film roles in sperm and semen [15], and amniotic fluid [16], while cholesterol esters of (O-acyl)-ω-hydroxy-fatty acids predominate in the vernix caseosa of fetal skin [23].

We have previously characterized anti-inflammatory very-long-chain dicarboxylic acids in a number of human biofluids [8]. This is the first report of these anti-inflammatory lipids in tear film fluid. In contrast to the (O-acyl)-ω-hydroxy-fatty acids and cyclic phosphatidic acids, these lipids were not augmented by the immune response to *Moraxella bovis*. This may be due to the final steps in the biosynthetic pathway of these lipids. As with (O-acyl)-ω-hydroxy-fatty acids, the first step involves fatty acid elongation via ELOVL4 and the subsequent ω-hydroxylation involves CYP4A/4F enzyme systems [24] with the subsequent conversion to a dicarboxylic acid by alcohol dehydrogenase and fatty aldehyde dehydrogenase [25]. Characterization of these enzyme systems in Meibomian glands remains to be undertaken.

Cyclic phosphatidic acids are generated by the transphosphatidylation of lysophosphatidylcholines by phospholipase D [14] (Figure 5). Lysophosphatidic acid, a lipid mediator that has the reciprocal pharmacodynamic profile of cyclic phosphatidic acids, is synthesized by hydrolysis of lysophosphatidylcholines by autotaxin and by phospholipase A2 hydrolysis of phosphatidic acids [14] (Figure 5). Augmentation of the anti-inflammatory actions of cyclic phosphatidic acids [6,7] by the immune response to *Moraxella bovis*, without increases in lysophosphatidic acids demonstrates the directed induction of key anti-inflammatory lipids.

The wealth of data from the Serhan laboratory regarding anti-inflammatory lipids generated from essential fatty acids has clearly established the roles of these lipids in immune resolution [5]. The eicosapentaenoic acid products, resolvin E1 and resolvin E2, are both potent anti-inflammatory lipids [4,10] but are undetectable in human emotional tears [26]. Navamepent (RX-10045) is a resolvin E1 analog that is currently under clinical development for the treatment of dry eye [27]. Resolvin E2 has been monitored in the plasma of healthy control humans [10] but our data are the first to demonstrate that resolvin E2 is present in the tear film of normal cattle and is dramatically induced by infection with *Moraxella bovis*. The induction of this lipid presumably can act in synergy with the anti-inflammatory action of cyclic phosphatidic acids. While other pro-resolving lipids were not detected in bovine tear film, it could well be the result of insufficient sensitivity or ion suppression with our direct infusion method.

*Moraxella bovis* possesses extensive lipolytic enzymes that might also contribute to the observed elevations in glycerophospholipid metabolites [28,29]. However, the selective augmentation of cyclic phosphatidic acids and not lysophosphatidic acids (Figure 5), suggests that the observed ocular lipid film changes represent a coordinated host response to the bacterial infection. Stimulation of the host immune response presumably results from bacterial secretion of lipooligosaccharide [30]. Alterations in the structural glycerophospholipids of the tear film also are presumably a host response. However, the lipid composition of the biofilm that is produced by *Moraxella bovis* [31] remains to be characterized to determine if this biofilm may contribute to the observed changes in structural lipids.

With regard to the augmented OAHFAs, these are unlikely to originate from *Moraxella bovis* since they are a novel class of amphiphilic lipids that are only present in ocular fluids [12,13,14,18,19,20,21,22] and several other unique compartments [15,16,23].

In summary, the current data suggests that the lipid alterations we monitored in cattle with pinkeye infections most likely represent a host response to the bacteria.

## 4. Materials and Methods

### 4.1. Clinical Samples

A herd of approximately 1200 Black Angus cattle of various ages (0–15 years) were used in this study. The cattle were visually evaluated in groups of 70–100 at least every 30 days for active Infectious Bovine Keratoconjunctivitis (IBK). When active IBK infection was visually noted, samples were collected for lipidomics analyses. A picture of the IBK affected eye(s) was captured, an intravenous blood sample was collected via the caudal tail vein, and a sample of ocular (tear) fluid was collected by placing two cotton tipped swabs in the lower eyelid. The cotton tipped swabs were placed in 1 mL of methanol and transported to the laboratory for analyses. Following sample collection, the affected animals were treated with tulathromycin or gamithromycin and an eye patch was applied to the severe cases. The pictures were evaluated to determine the Stage of the pinkeye infection as described by W.D. Whittier, et al. (https://web.extension.illinois.edu/oardc/downloads/43489.pdf). The veterinarian collecting the samples also staged the disease in each animal for this study. The control group consisted of nine 9-month, four 1-year, and one 2-year female cattle while the pink eye group consisted of six 1.5-year, three 6 year, two 10 year, and one 11-year cattle. The stages of pinkeye were I (N = 2), II (N = 2), III (N = 4), and IV (N = 4). The Lincoln Memorial University IRB approval for this study was 1809-CLIN.

### 4.2. Lipidomics

To the cotton swabs in 1 mL of methanol were added stable isotope internal standards, 1 mL of water and 2 mL of methyl-tert-butyl ether. The tubes were next vigorously shaken at room temperature for 30 min prior to centrifugation at 4000× *g* for 30 min at room temp. The upper organic layer was isolated and dried by centrifugal vacuum evaporation and dissolved in isopropanol:methanol:chloroform (4:2:1) containing 7 mM ammonium acetate. The stable isotope internal standards included [^2^H_5_]PtdE 34:1, [^2^H_5_]DHA, [^2^H_31_]PtdC 34:1, [^2^H_28_]DCA 16:0, [^2^H_7_]cholesterol sulfate, and bromocriptine as internal standards [15,16,32].

Direct infusion lipidomics utilized high-resolution data acquisition, with an orbitrap mass spectrometer (Thermo Q Exactive). In negative ion electrospray ionization (ESI), the anions of ethanolamine plasmalogens (PlsE), (O-acyl)-ω-hydroxy-fatty acids (OAHFA), cyclic phosphatidic acids (cPA), phosphatidylethanolamines (PtdE), lysophosphoethanolamines (LPE), lysophosphatidic acids (LPA), sphingosine 1-phosphate (S-1-P), phosphatidylglycerols (PG), phosphatidylinositols (PI), and phosphatidylserines (PS). In positive ion ESI, the cations of choline plasmalogens (PlsC), phosphatidylcholines (PtdC), lysophosphatidylcholines, and sphingomyelins were quantitated utilizing high-resolution mass spectrometry (<2 ppm mass error). The cations and anions of bromocriptine were used to monitor for potential mass axis drift. Between injections, the transfer line was washed with successive 500 µL washes of methanol and hexane/ethyl acetate/chloroform (3:2:1).

For NESI MS^2^ experiments of potential anti-inflammatory mediators, the following transitions (Precursor → Product) were monitored [26,28]: RvE2 (333.2 → 253.1234); RvE3 (333.2 → 201.1648); RvE1 (349.2 → 195.1026); RvD5 (359.2 → 199.1492); RvD6 (359.2 → 101.0244); Mar1&2 (359.2 → 221.1183); PD1 (359.2 → 153.0921); RvD1 (375.2 → 203.1441); RvD2 (375.2 → 247.1339); RvD3 (375.2 → 153.0921); OH-Mar1 (375.2 → 221.1183); OH-PD1 (375.2 → 153.0921); COOH-Mar1 (389.2 → 221.1183); COOH-PD1 (389.2 → 153.0921); cPA 16:0 (391.2 → 255.2329); cPA 18:0 (419.3 → 283.2642); VLCDCA 28:4 (445.3 → 383.3319).

### 4.3. Statistical Analysis

Since the samples were absorbed onto cotton swabs for extraction, we had no index of volume collected. Therefore, standardization of the data based on tear volume was not possible. However, there were no significant differences in the ion intensities of internal standards between samples, and the RSDs for raw peak areas were less than 50%, comparable to previous infection studies where lipid metabolites were normalized to the volume of the analyte [17]. Therefore, semi-quantitative data are presented as the raw peak areas for each of the high-resolution masses. The data presented in graphs are from the pilot study with 12 controls and 12 cattle with pink eye. Data from a validation study of 12 controls and 27 infected cattle yielded identical results. A Student’s t-test (Microsoft Excel), assuming equal or unequal variances, was used to determine significant differences in levels of metabolites between pink eye animals and controls, subsequent to an F-test (Microsoft Excel) to determine if the variances between groups were statistically different.

## 5. Conclusions

The immune response to ocular infections with *Moraxella bovis* results in the induction of endogenous anti-inflammatory lipids including resolvin E2 and cyclic phosphatidic acids. Infection also augments the levels of structural glycerophospholipids and sphingolipids and critical amphiphilic lipids essential to the lipid tear film layer. This included (O-acyl)-ω-hydroxy-fatty acids and cholesterol sulfate. Increased levels of hydroperoxy glycerophospholipids also indicate that this bacterial infection results in lipid peroxidation. Our study also reports for the first time the presence of cyclic phosphatidic acids and very-long-chain dicarboxylic acids in tear film fluid.

## Figures and Tables

**Figure 1 metabolites-08-00081-f001:**
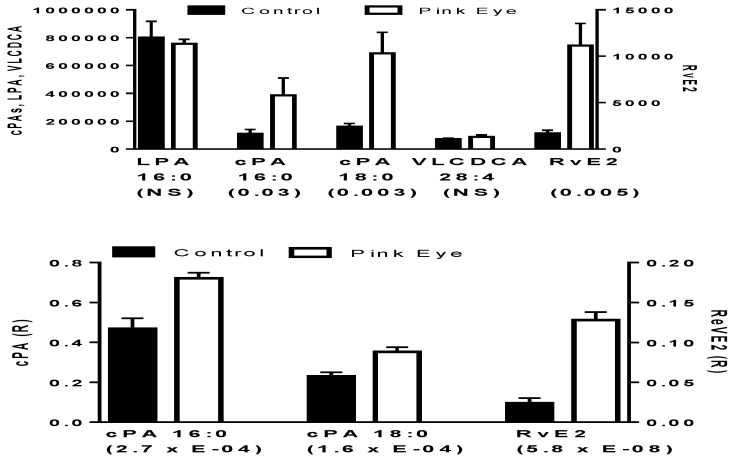
Potential anti-inflammatory biomarkers in bovine tear fluid. In the upper graph data are presented as peak intensities for the product ions (MS^2^) of each of the lipids ± SEM. In the lower graph, product ion data are presented as the ratios (R) of the endogenous lipid peak intensity to the peak intensity of 500 pmoles of [^2^H_4_]cPA 16:0. cPA, cyclic phosphatidic acid; LPA, lysophosphatidic acid; RvE2, resolvin E2; VLCDCA, very-long-chain dicarboxylic acid. *p* values are in brackets.

**Figure 2 metabolites-08-00081-f002:**
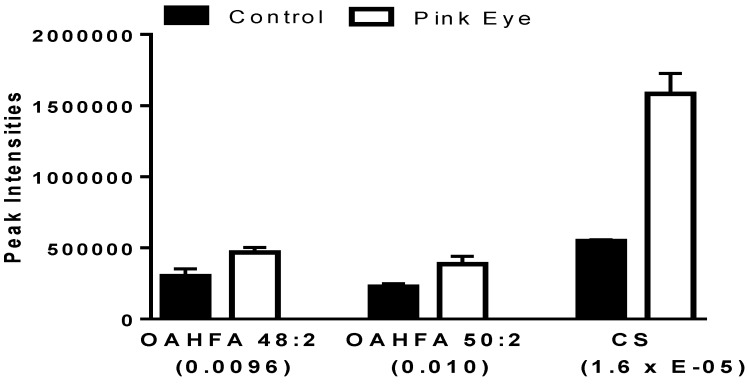
Charged anionic lipids in bovine tear fluid. Data are presented as peak intensities for the ions of each of the lipids ± SEM. OAHFA, (O-acyl)-ω-hydroxy-fatty acid; CS, cholesterol sulfate. *p* values are in brackets.

**Figure 3 metabolites-08-00081-f003:**
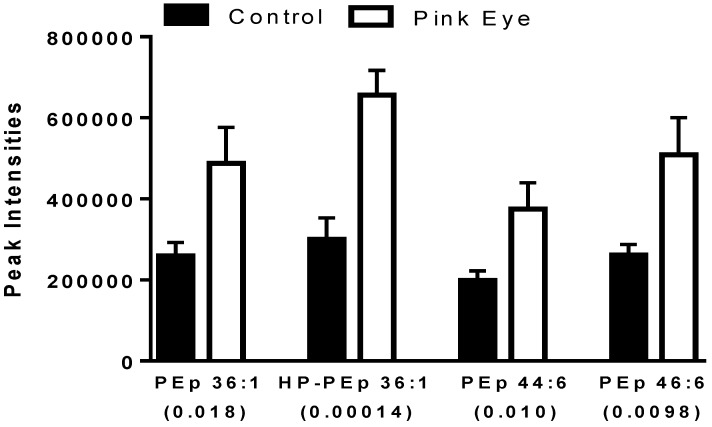
Ethanolamine plasmalogens in bovine tear fluid. Data are presented as peak intensities for the ions of each of the lipids ± SEM. HP, hydroperoxy; PEp, ethanolamine plasmalogen. *p* values are in brackets.

**Figure 4 metabolites-08-00081-f004:**
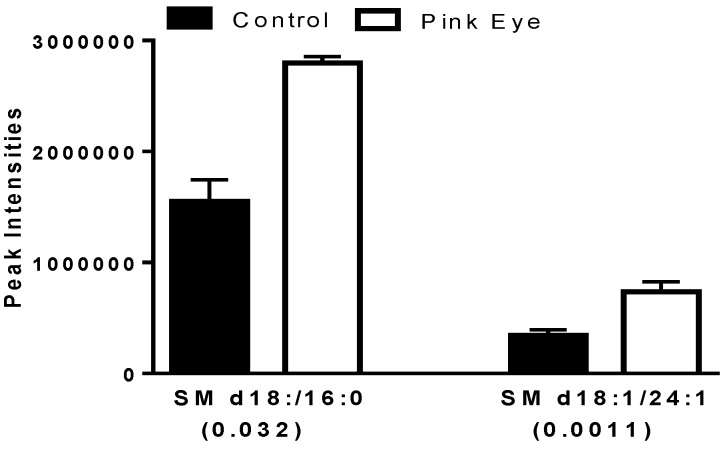
Sphingomyelins in bovine tear fluid. Data are presented as peak intensities for the ions of each of the lipids ± SEM. SM, sphingomyelin. *p* values are in brackets.

**Figure 5 metabolites-08-00081-f005:**
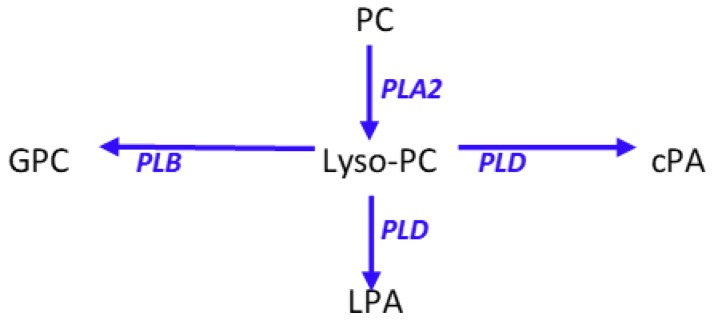
Schematic presentation of phosphatidylcholine (PC) metabolism to lysophosphatidylcholines (lyso-PC) and the associated metabolites cyclic phosphatidic acid (cPA), lysophosphatidic acid (LPA), and glycerophosphocholine (GPC). PLA2, phospholipase A2; PLB, phospholipase B; PLD, phospholipase D.

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
