# Peer review of "Tear Film Amphiphilic and Anti-Inflammatory Lipids in Bovine Pink Eye"

_metabolites, 2018, doi:10.3390/metabo8040081_

Round 1

Reviewer 1 Report

The present manuscript describes an interesting approach to analyze "Tear Film Amphiphilic and Anti-inflammatory Lipids in Bovine Pink Eye". Although the results and conclusions are sound and the overall topic of interest to the readers of metabolites there is potential for a revision before acceptance of the contribution.

The analysis of E-resolvines is controversially discussed in the literature. The biological existence is not clear; although their potential role in anti-inflammatory processes seems to be indisputable. Do the authors unambiguously identify these compounds in the biological matrix? I did not find the raw data showing these. It is still unclear if these compounds are of endogenous origin.

On page 7 the authors write "lysophosphatidylcholines, and sphingomyelins were quantitated". How was this done?

If the animals show signs of infection in the eye why there are not pro-inflammatoric eicosanoides detectable? Did the analysis show any signs of such compounds?

Overall the discussion part is not deep enough in my view: there is more potential to address infections related importance of the findings.

A major problem I see in the missing normalization: of course it is difficult to measure the sample amount. Did the authors tried to balance the swaps? Or maybe a normalization based on protein amount is possible?

One topic I like to address is the correlation between sex, age and infection outcome. Is there any possible correlation in the data?

minor points: some typos:

page 1: infectois bvine keratoconjunctivosous (IBK)

page 6: in 1 mL of metahanol

Overall the manscript can be improved by a more critical discussion of the resolving part and maybe some more efforts towards the normalization of the data.

Author Response

The E resolvins are difficult to measure due to their low levels. However there are a number of labs that have successfully monitored these lipids. In our case high-resolution tandem mass spectrometry unambiguously monitored resolving E2 in ocular fluid.

The quantitation of LPC and SM has been added to page 7.

The study design was to focus on anti-inflammatory and structural lipids not on inflammatory lipids or cytokines.

The discussion has been expanded and Fig. 5 added to organize the findings. This included the addition of 3 more references.

Normalization was an issue but the RSDs were no different from our previous studies of bacterial infections on plasma lipids where we could measure the volume. This statement has been added to the methods.

The ages and infection status has been added to the methods and the results sections.

The IBK spelling has been corrected on page 1.

Methanol spelling has been corrected on page 6.

All changes have been highlighted in RED.

Reviewer 2 Report

Wood and co-authors in the manuscript “Tear Film Amphiphilic and Anti-inflammatory Lipids in Bovine Pink Eye” deal with alterations of tear film lipidome under conditions of the immune response. This is a very interesting, timely, and important topic as lipids constitute a key component of the tear film, and changes of lipidome are well known to influence tear film properties strongly. High-resolution mass spectrometry is employed in bovine pink eye tear samples. The authors focus on both anti-inflammatory lipids and “standard” structural tear film lipids in the bovine pink eye”. The study finds a dramatic increase of the anti-inflammatory lipids resolvin E2 and cyclic phosphatidic acids. Moreover amounts of some of the structural lipids were also enhanced, in particular (O-acyl)-ω-hydroxy-fatty acids and sphingomyelins. Products of lipid peroxidation were also found. Notably, the presence of cyclic phosphatidic acids and very-long-chain dicarboxylic acids is reported for first time in the tear film.

This is a solid and important work regarding tear film lipidome. Therefore, I recommend it for publication in the Metabolites in the current form.

Author Response

This reviewer had no issues.

Reviewer 3 Report

The authors present interesting results clearly demonstrating by MS changes in the tear film lipidom in the inflamed eye. I have only one concern: in the abstract the authors claim that "dramatic elevations" of anti-inflammatory lipids are found, which is true and which is an important finding. At the end of the introduction they say, that their study is preliminary. I would say it is a first but important step in the analysis of the tear film lipidom under infection.

Author Response

The statement has been modified as suggested.

Round 2

Reviewer 1 Report

The authors have addressed the issues I raised in my review in a sufficient way.

The manuscript is acceptable after editorial spelling check.